# Microsatellite Characterization and Panel Selection for Brown Bear (*Ursus arctos*) Population Assessment

**DOI:** 10.3390/genes13112164

**Published:** 2022-11-19

**Authors:** Vincenzo Buono, Salvatore Burgio, Nicole Macrì, Giovanni Catania, Heidi C. Hauffe, Nadia Mucci, Francesca Davoli

**Affiliations:** 1Unit for Conservation Genetics (BIO-CGE), Department for the Monitoring and Protection of the Environment and for Biodiversity Conservation, Italian Institute for Environmental Protection and Research (ISPRA), Ozzano dell’Emilia, 40064 Bologna, Italy; 2Department of Biological, Geological and Environmental Sciences (BiGeA), University of Bologna, 40100 Bologna, Italy; 3Conservation Genomics Research Unit, Research and Innovation Centre, Fondazione E. Mach, San Michele all’Adige, 38098 Trento, Italy

**Keywords:** fragment analysis, genetic diversity, individual identification, invasive and non-invasive genetic profiles, marker selection, *Ursus arctos marsicanus*, *Ursus arctos arctos*

## Abstract

An assessment of the genetic diversity and structure of a population is essential for designing recovery plans for threatened species. Italy hosts two brown bear populations, *Ursus arctos marsicanus* (*Uam*), endemic to the Apennines of central Italy, and *Ursus arctos arctos* (*Uaa*), in the Italian Alps. Both populations are endangered and occasionally involved in human–wildlife conflict; thus, detailed management plans have been in place for several decades, including genetic monitoring. Here, we propose a simple cost-effective microsatellite-based protocol for the management of populations with low genetic variation. We sampled 22 *Uam* and 22 *Uaa* individuals and analyzed a total of 32 microsatellite loci in order to evaluate their applicability in individual identification. Based on genetic variability estimates, we compared data from four different STR marker sets, to evaluate the optimal settings in long-term monitoring projects. Allelic richness and gene diversity were the highest for the *Uaa* population, whereas depleted genetic variability was noted for the *Uam* population, which should be regarded as a conservation priority. Our results identified the most effective STR sets for the estimation of genetic diversity and individual discrimination in *Uam* (9 loci, PIC 0.45; PID 2.0 × 10^−5^), and *Uaa* (12 loci, PIC 0.64; PID 6.9 × 10^−11^) populations, which can easily be utilized by smaller laboratories to support local governments in regular population monitoring. The method we proposed to select the most variable markers could be adopted for the genetic characterization of other small and isolated populations.

## 1. Introduction

Conservation genetics has proved to be effective in evaluating species viability by providing estimates of heterozygosity and genetic diversity of populations [1,2,3,4]. In addition, by identifying individual genotypes, it is possible to quantify the minimum number of individuals, estimate the effective population size, and infer genetic structures, as well as reconstruct kinship and parentage, and demographic processes [5,6,7,8,9,10]. Non-invasive genetic methods (those relying exclusively on environmental DNA—eDNA—from feces, hair, saliva, etc.) are commonly employed in elusive and rare species management, especially in carnivore populations [9,11,12,13,14], so that populations can be studied without physically capturing and handling individuals, thereby reducing behavioral responses and risks of trap-related injuries [11,15].

Population genetic characterization can be performed using different types of genetic markers, but microsatellite loci, or short tandem repeats (STRs), have been widely adopted in non-invasive genetic monitoring, as they are relatively cheap to characterize, and can be reliably amplified from eDNA. STRs have played an important role in ecological, evolutionary, and conservation research in recent decades, as they are particularly informative [16,17]. For example, they are suitable genetic markers [18] for gene flow analyses [19], genetic diversity [20,21,22,23], paternity testing, population genetics [24], and kinship studies [25,26,27]. Such baseline information is important for defining and evaluating management strategies, and identifying populations undergoing inbreeding depression or those sufficiently differentiated to warrant management as an evolutionarily significant unit (ESU).

Italy is inhabited by two isolated bear populations, *Ursus arctos marsicanus* (*Uam*, Apennine brown bear, Figure 1a) and *Ursus arctos arctos* (*Uaa*, Alpine brown bear, Figure 1b). The future of both populations is still rather uncertain. *Uam* is a critically endangered endemic taxon under IUCN criteria, while the *Uaa* population is the result of a translocation of 10 individuals from Slovenia to the Province of Trento, Italy, aimed at rescuing autochthonous individuals from extinction. In the Apennines, a few dozen *Uam*, representative of a unique genetic clade, are still under threat from human activities [28], while in the Alps, the reintroduction of *Uaa* 20 years ago prevented their extinction [5,29]; many reproduction events have been recorded, with newborns contributing to the population increase [30].

The 20-year monitoring projects of both brown bear populations in Italy are currently using 13 and 15 STRs for individual identification in the *Uam* [33] and *Uaa* [34] populations, respectively. However, the low genetic variability in *Uam* and the low discrimination power of the current microsatellite panel will almost certainly reduce the resolution of individual genotyping in the long term. Moreover, genotyping errors due to a high incidence of false alleles and allelic drop out could lead to misidentified genotypes [35]. Therefore, an improved set of highly variable markers is needed to reduce the uncertainty of genotyping, and to enable reliable individual brown bear identification over time. In addition, for *Uaa*, the implementation of a standardized method would allow the comparison of data produced by different cross-border labs across the range of the subspecies, promoting more effective conservation actions.

The present study is focused on using variability indices to develop an objective method for ranking polymorphic STR markers, choosing the panel of these markers with the highest statistical power in individual identification and genotype reconstruction, and testing the efficiency of these panels for small populations. The ultimate goal is to design efficient marker panels that can be used for inexpensive genotyping for the management of threatened species. Here, using traditional capillary electrophoresis and invasive and non-invasive samples of *Uam* and *Uaa*, we tested the performance of the following four different STR marker sets: (i) the loci commonly used by Italian laboratories (CURRENT, 13 loci in *Uam* and 15 in *Uaa*); (ii) a new set of 13 STRs, originally developed for high-throughput sequencing (HTS) for the large Pyrenean and Dinaric brown bear populations [36] (NEW); (iii) the complete set of loci from (i) and (ii) (TOT), and (iv) the subset of loci for both the *Uam* and *Uaa* populations with the highest resolution power for individual identification, i.e., the optimal set of STR markers to use in long-term monitoring projects (BEST).

## 2. Materials and Methods

### 2.1. Sample Selection

The NEW panel was tested using traditional capillary electrophoresis on a total of 44 different wild bears, 22 from the *Uam* population, Central Italy (Figure 1a), and 22 from the *Uaa* population, Northern Italy (Figure 1b). Biological samples comprised 17 invasive (blood and tissues from individuals live-captured and released for research purposes, or from bear carcasses), and 27 non-invasive (hair and feces deposited by wild bears in the environment) samples, collected during population surveys (Table 1). We chose both invasive and non-invasive biological samples to test the efficacy of the markers on both types of DNA. Most individuals in our study were systematically or opportunistically sampled from wild brown bear populations using non-invasive methods (Table 1). The same 44 Italian wild bears were genotyped with the full set of 32 microsatellite loci. The sex of all individuals was determined by sequencing the amelogenin gene.

The sampling locations were located on public land, and no specific permission was required for sample collection. In addition, the animals described herein are not considered experimental animals as defined by the EU directive 2010/63. Consequently, we were not required to seek ethical review or approval of this study. We applied the principles of non-disturbance of animals and the environment. A large majority of the individual bears included in our study were sampled non-invasively, i.e., remotely, by obtaining biological samples (hairs; feces) without the need to capture and handle individual animals. A minority of the bears were found dead, and their tissue samples were dissected from the carcasses, whereas the remaining bears were live-trapped for research purposes [37,38] and released on site after handling. Permits for bear capture were issued by the Italian Ministry of the Environment (permit numbers: m_amte.PNM.REGISTRO UFFICIALE.U.5344.17-03-2014; 6270.28-03-2017; 12959.14-06-2018).

### 2.2. Microsatellite Genotyping and Analysis

All genetic analyses were carried out in the Unit for Conservation Genetics (BIO-CGE) at ISPRA, or the Animal, Environmental and Antique DNA Platform at the Fondazione E. Mach, using the same harmonized protocols (Appendix A). Total genomic DNA was isolated from invasive and non-invasive samples with the QIAGEN DNeasy^®^ Blood & Tissue Kit (QIAGEN, Hilden, Germany), using a robotic workstation for automated purification of DNA (QIAcube, QIAGEN, Hilden, Germany). Since the DNA obtained from non-invasive sampling is often diluted and deteriorated, a multiple-tubes approach, with positive and negative controls at each step, was used to prevent stochastic errors [33,34,39]. Multilocus genotypes were obtained by evaluating the results of four independent PCR replicates, following the protocols described in [33]. The microsatellite markers of the NEW panel were amplified in three multiplex PCR systems (Table 2). A universal tag was added in the 5′-terminus of each forward primer. Tags were fluorescently labelled with FAM, HEX, NED, or PET dyes (Applied Biosystems, Foster City, CA, USA). PCR conditions were optimized with the QIAGEN^®^ Multiplex PCR Kit (QIAGEN, Hilden, Germany) in Mastercycler^®^ pro S (Eppendorf SE, Hamburg, Germany) and Veriti™ 96-Well Thermal Cycler (Applied Biosystems™). PCR reactions were performed in a total volume of 8 µL, which comprised the following: 3.5 µL of 2× QIAGEN Multiplex PCR Master Mix (mixture of Multiplex PCR buffer, dNTPs, and Hot Start Taq DNA polymerase, providing a final concentration of 3 mM MgCl2), 0.7 µL of 5× Q-solution (an additive that enables efficient amplification of difficult templates, e.g., GC rich), a total volume equal to the sum of individual primer volumes of the fluorescently labelled primer mix (10 μM), 2.0 µL of 10–100 ng/µL DNA template, and adjusting volume to 8 μl with RNase-free water. We adopted the following thermal cycling conditions: initial denaturation for 15 min at 95 °C, for 45 cycles; 30 s at 95 °C, 90 s at the annealing temperature of the primer mix (Ta = 57 °C), 60 s at 72 °C and 10 min at 72 °C for final elongation. STR fragments were detected and sized using capillary electrophoresis on an ABI Prism 3130XL Genetic Analyzer DNA sequencer (Thermo Fisher Scientific, Waltham, MA, USA) at the ISPRA BIO-CGE lab. The electropherograms were collected by the Data Collection Software v.3.0. GeneScan-500 LIZ and GeneMapper^®^ Software v. 4.1 (Applied Biosystems by Thermo Fisher Scientific) were used to calibrate and determine allelic size, respectively.

MicroChecker 2.2.3 [40] was used to evaluate null alleles with an attuned *p*-value, after Bonferroni correction, conforming to α = 0.05 [41]. False alleles (FA) and rates of allelic dropout (ADO) were estimated using GIMLET v1.3.3 [42]. The reliability of each genotype was determined using RelioType [43], with a confidence level of 95%, following the procedure illustrated in [33]. GenAlEx v6.51b2 [44,45,46] was used to assess genetic diversity analysis measured as allele frequencies, the mean number of alleles per locus (Na), effective numbers of alleles (Ne), Shannon’s information index (I), observed (Ho) and expected unbiased (UHe) heterozygosity, Hardy–Weinberg equilibrium (HWE), and probability of identity for unrelated individuals (PID) and for siblings (PIDsib). The expected heterozygosity (He) is usually used to describe genetic diversity, as this index is less sensitive to the sample size than the observed heterozygosity (Ho). Sequential Bonferroni correction was used to adjust the *p*-values (corresponding to 0.001) in multiple tests [41]. MM-Dist [47] was used for estimating the distribution of genotypic differences (mismatches) between individuals in both populations.

The power of individual markers to discriminate individual multilocus genotypes [48,49] was evaluated considering the polymorphism information content (PIC, PowerMarker 3.25, [50]), jointly with the probability of identity for pairs of siblings for a given locus (PIDsib), and with the genotyping error rates (ADO and FA, GIMLET v1.3.3). For each bear population, the most informative markers (BEST panels) were selected by choosing those that simultaneously showed PIC > 0.35, ADO < 0.10, FA = 0, and PID < 0.40.

To evaluate marker informativeness and select the optimal set of STR markers to be used for long-term genetic monitoring, we compared parameters of the following four different STR marker sets: CURRENT (13 loci in *Uam* and 15 in *Uaa*), NEW (13 loci from high-throughput sequencing for both populations), TOT (26 in *Uam*; 28 in *Uaa*), and BEST, including the probability of identity for the increasing number of loci (PID, [51]); the equivalent probability for pairs of siblings (PIDsib, [49]); the number of mismatches observed between pairs of different genotypes that matched at all loci (0-MM) and at all loci but one, two, and three loci (1-MM, 2-MM, and 3-MM pairs, [52]); and the marker index (a statistical parameter used to estimate the total utility of the maker set). The marker index (MI) was the product of the PIC and effective multiplex ratio (EMR) [53,54], i.e., the higher the MI, the higher the informativeness. The EMR for codominant loci was calculated as the proportion of the total number of effective alleles (per primer) per total number of primers [53,54], so that the higher the value of EMR, the more efficient the marker set is. The PID and PIDsib values were computed for one-to-all loci, by adding loci sequentially, from the highest to the lowest level of informativeness, based on the expected number of different individuals with the same genotype at a given locus.

## 3. Results

We successfully genotyped for the NEW marker set 44 DNA samples (8 from blood, 6 from tissue/bones, 28 from hair, and 2 from feces; Table 1), selected from the Italian national biobank provided by ISPRA BIO-CGE. These 44 DNA samples belonged to 22 different wild bears from the *Uam* population and 22 from the *Uaa* population (20 females; 24 males; Table 1). All markers were amplified with an overall genotyping success rate of 100% in *Uam* and 99% in *Uaa*. All 13 loci were polymorphic in *Uaa*, whereas 3 loci were monomorphic in *Uam* (Table 2). No markers deviated significantly from HWE after Bonferroni correction. The allelic richness was the highest in *Uaa* (Ne = 1.587 ± 0.188 in *Uam*, 2.450 ± 0.254 in *Uaa*). UHe was 0.270 (SE ± 0.074; range: 0–0.644) and 0.561 (SE ± 0.039; range: 0.333–0.816) for *Uam* and *Uaa*, respectively. There was no evidence of the presence of null alleles or allelic dropout in either *Uam* or *Uaa* populations. Genotyping errors were found in six loci in *Uam* and in four loci in *Uaa* (Table 2). Invasive samples (11 *Uam* and 6 *Uaa*) produced particularly reliable genotypes with one ADO (rate: 0.24%) and two FA (rate: 0.49%) at one locus. In non-invasive samples (6 *Uam* and 17 *Uaa*), 28 ADO (rates: 1.9–12.1%, mean rate: 3.1%) and no FA were observed at 7 loci. The PID and PIDsib values for the NEW set of STRs showed that the most informative loci were UA25 (PID = 0.20; PIDsib = 0.48) in *Uam*, and UA16 (PID = 0.07; PIDsib = 0.37) in *Uaa* (Table 2). The mismatch distribution estimated by MM-DIST revealed that the frequencies of two individuals that differed (mismatching) by 0 to 13 loci were at their highest at 6 loci (0.290) in *Uam* and at 11 loci (0.247) in *Uaa*, but even then, most of the genotypes differed by more than 3 loci. Hence, the MM distribution suggested that the *Uaa* population was more variable than the *Uam* population because most individuals within the populations differed by 11 and 6 loci, respectively (Figure 2a,b).

Polymorphism and genotyping errors of the CURRENT panel were extrapolated from [33] (*Uam*) and [34] (*Uaa*) (Table 3).

For the TOT set, the gene diversity (GD) was 0.40 and 0.61 for *Uam* and *Uaa*, respectively (Table 3). A total of 5 out of 26 loci in *Uam* and 20 out of 28 loci in *Uaa* were highly polymorphic (PIC ≥ 0.5).

Based on the comparative values of PIC, error rates, number of mismatches and PID across both populations, we selected a panel of 9 and 12 microsatellite markers for *Uam* and *Uaa*, respectively (BEST, Table 4), which were suitable for the genetic monitoring of these populations. These included five di-nucleotide (*Uam* and *Uaa*) and four (*Uam*) or seven (*Uaa*) tetra-nucleotide microsatellites. The mean PIC for selected loci was 0.45 and 0.64 for *Uam* and *Uaa*, respectively (Table 3). The cumulative power of discrimination among unrelated individuals (PID) using the selected panel was 2.0 × 10^−05^ and 6.9 × 10^−11^ for *Uam* and *Uaa*, respectively. Similarly, using the derived panel (BEST), the cumulative power of discrimination among siblings (PIDsib) was 5.3 × 10^−03^ and 5.4 × 10^−05^ for *Uam* and *Uaa*, respectively.

Among the four STR marker sets (CURRENT, NEW, TOT and BEST), the highest values of the genetic diversity indices (UHe, GD and PIC) were reported for the BEST STR panel in both populations (Table 3 and Table 4). The resolution power of the different sets is shown in Figure 3a,b and Table 4. Although in *Uam,* the BEST set of loci showed a weak signal of mismatches among genotypes (2MM = 1.65 × 10^−02^; 3MM = 1.86 × 10^−02^), this set demonstrated higher MI values for both *Uam* and *Uaa* populations (Table 4).

## 4. Discussion

To reduce the dimensionality of genetic data and provide the smallest possible number of loci for optimal identification of individual multilocus genotypes, it is important to construct a set of appropriate markers. Marker sets are distinguished by the extent (i.e., magnitude) of their informativeness, in turn depending on the degree of polymorphism. Many studies have used microsatellite panels for addressing issues related to endangered species [33,62,63,64]. However, up to now, STR marker sets have been chosen based on the presence of polymorphism in each marker, with no attempt at optimization. This means that over time, in small populations, an ever-increasing number of markers is needed to identify individuals with a high degree of probability. Instead, our proposed panels are pioneering attempts to reduce costs and laboratory efforts. Moreover, the proposed selection method, by detecting the most variable markers, has the potential to be adopted for the genetic characterization of small and isolated populations of different taxa, improving the reliability/time/cost trade-off of the genetic analysis.

To select the most appropriate marker set, we used not only PID or heterozygosity values, but also other parameters, including allelic richness, gene diversity, distribution of allelic frequencies, presence of PCR errors (that, if not corrected, would lead to an over- or underestimation of the number of individuals), and indices such as PIC and MI. By applying this method to brown bear populations in Italy, we identified the BEST sets, reducing the number of markers (from 13 to 9 in *Uam*, from 15 to 12 in *Uaa*), decreasing the risk of genotyping errors (ADO 0.04 vs. 0.05 in *Uam*, 0.02 vs. 0.08 in *Uaa*; FA 0 vs. 0.01 in *Uam* and 0.03 in *Uaa*), and lowering the cost of genotyping.

Considering the total number of individuals estimated for small, isolated brown bear populations in Italy [65], the proposed BEST panels of 9–12 microsatellites provided sufficient evidence of polymorphism to undertake genetic research aimed at establishing genetic identification. For example, according to [51], a minimum PID of 0.001 is required to distinguish between individuals, while a minimum PIDsib of 0.05 is required to distinguish between siblings (see also [33,66]). This value of PID was sufficiently low to discriminate between individuals accurately, since the expected population size was not greater than a few hundred individuals [51,66,67,68]. Woods et al. [51] reported that in brown bears, for instance, 4–6 microsatellites were sufficient to accurately distinguish individuals and siblings. The low values of PID and PIDsib obtained for five loci suggest that individuals can be identified using a low number of loci, ranging between five and seven [51,66,67]. Instead, for the much smaller *Uam* and *Uaa* populations, 9 and 12 microsatellite loci are sufficient to identify individuals (i.e., the values of PID and PIDsib were below the above thresholds; Table 3 and Table 4). Thus, the BEST panel comprises species-specific markers that avoid cross-species amplification; these validated panels proved to be more informative and reliable than the STR sets currently used and also demonstrated an improved discrimination capacity, increasing the probability of individual identification, while reducing PID and shadow effects.

We also show here that the traditional method currently used for STR genotyping, fragment analysis by capillary electrophoresis, was effective for microsatellite loci developed for high-throughput sequencing (NEW set), probably because the shorter markers (<210 bp) improved the amplification of degraded DNA. This is important, since HTS only becomes cost-effective when many samples are analyzed simultaneously. For smaller populations, this is not always possible, or desirable (for example, many human–bear conflicts require genotyping of very few samples within a few days to quickly resolve indemnity issues). To further reduce costs, the proposed BEST set of STRs could be optimized into multiplex PCRs. The use of markers with overlapping amplification products (total observed allele size range for the BEST set of markers: 88–132 bp in *Uam* and 78–203 bp in *Uaa*) and the selection of a panel with compatible primer properties (Dye Set G5) also contributed to maximizing the amplification performance in multiplex PCRs by reducing bias between markers. In addition, the preferential use of tetranucleotide repeats allowed a reduction in the number of PCR-induced stutter sequences in the outputs of amplified microsatellites [69], facilitating the allele reads compared to dinucleotide repeats.

Previous studies have shown that on the one hand, estimates of heterozygosity based on a few loci provide poor estimates of genome-wide genetic variation, and may not allow the differentiation of individuals, potentially leading to the underestimation of population size [51]. On the other hand, a high number of loci can also have a negative impact on individual identification [70]. Our results confirmed these conclusions. For example, we estimated a total proportion of error rates (ER) of 1.43 (ADO 1.33; FA 0.10) in *Uam* for 17 microsatellite loci, corresponding to a mean ER proportion of 0.084 per locus (TOT set), and an ER of 1.78 (ADO 1.32, FA 0.46) in *Uaa* for 19 loci, corresponding to a mean ER of 0.094 per locus (TOT set). Increasing the number of loci may decrease ER, generating false unique genotypes, and leading to an overestimated population size [70].

This study successfully identified optimal STR panels that perform reliably using invasive and non-invasive samples, which are useful for monitoring and protecting threatened brown bear populations in the future. The adopted workflow could be replicated in any small population affected by low genetic variability for which reliable and effective long-term genetic monitoring could be helpful to detect changes in patterns of variability, or to confirm the effectiveness of management practices. On the IUCN Red List of threatened species, the brown bear (*U. arctos*) is currently listed at a global level as ‘Least Concern’, but there are many small, isolated populations that are categorized as ‘Critically Endangered’; for example, both Italian populations have been placed on the Italian vulnerable species list (Criterion D). In fact, we estimated relatively low genetic diversity in the studied Italian bear populations, especially with regard to *Uam*, with three loci being monomorphic. The *Uam* population was characterized by an even lower genetic diversity compared to *Uaa* (uHe = 0.368 vs. 0.627; GD = 0.40 vs. 0.61), confirming the results reported in [33,34]. Since the management of both these small populations depends on regular genetic analyses, the BEST set of identified markers will be critical to keeping the cost-benefit ratio in favor of continued monitoring. Nevertheless, since small populations can be affected by genetic drift effects, marker panels should be regularly tested to verify their reliability over time and generations.

## Figures and Tables

**Figure 1 genes-13-02164-f001:**
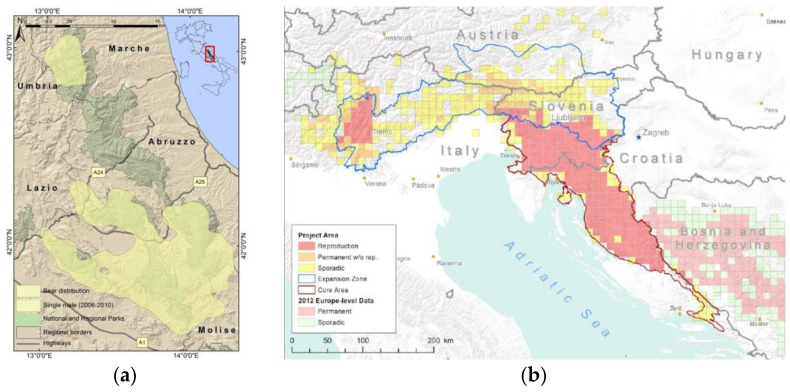
Brown bear (*U. arctos*) distribution in Italy and neighboring countries: (**a**) brown bear distribution in the Central Apennines (*Uam* population), 2005–2014 (from [31]), the map of Italy provides a location reference at the country level (red frame); the main map provides a reference at the local level (Central Apennines); (**b**) brown bear distribution in the Alps (*Uaa* population) and neighboring countries (from [32]).

**Figure 2 genes-13-02164-f002:**
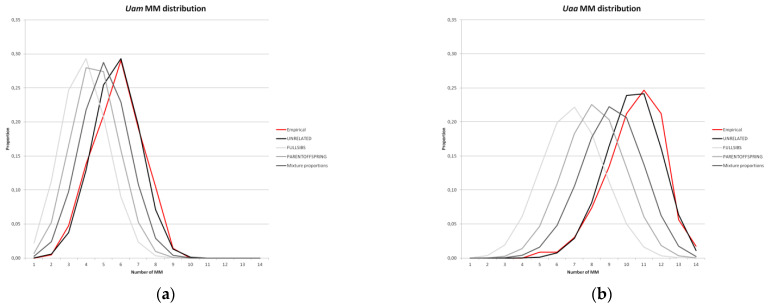
Mismatch probability distributions in Italian brown bear populations. Results are shown for the NEW STR set; the X-axis reports the number of mismatches and the Y-axis shows the probability: (**a**) *U. a. marsicanus* (*Uam*) population; (**b**) *U. a. arctos* (*Uaa*) population.

**Figure 3 genes-13-02164-f003:**
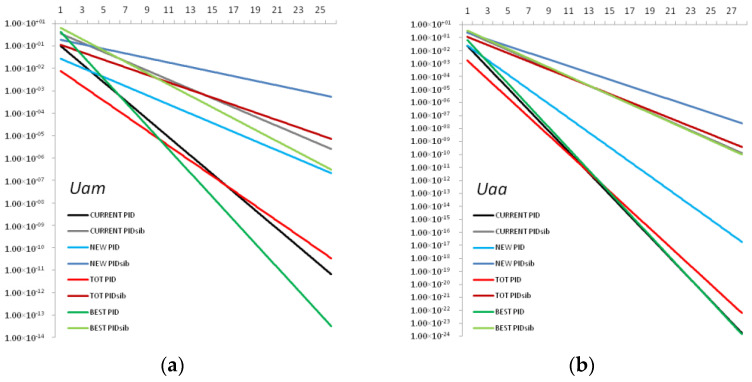
The theoretical predictions (trend lines) of the probability of identity for unrelated (PID) and related (PIDsib) individuals calculated for four different STR marker sets (CURRENT, NEW, TOT and BEST) for increasing locus combinations (the X-axis). Loci were combined from the most to the least informative for the (**a**) *U. a. marsicanus* (*Uam*) population; (**b**) *U. a. arctos* (*Uaa*) population.

**Table 1 genes-13-02164-t001:** Sample information.

Subspecies (Abbreviation)	Biological Sample	Sampling Type	Sampling Method	Sex	Genotype ID
*Ursus arctos*	Blood	Invasive	Capture and release	F	73
*marsicanus* (*Uam*)	Feces	Non-invasive	Opportunistic	M	149
	Blood	Invasive	Capture and release	F	31
	Hair	Non-invasive	Opportunistic	M	81
	Blood	Invasive	Capture and release	F	99
	Hair	Non-invasive	Opportunistic	M	127
	Hair	Non-invasive	Opportunistic	M	128
	Hair	Non-invasive	Opportunistic	M	135
	Hair	Non-invasive	Opportunistic	M	142
	Hair	Non-invasive	Rub tree	M	150
	Hair	Non-invasive	Rub tree	M	151
	Hair	Non-invasive	Damage	M	164
	Tissue	Invasive	Carcass	F	101
	Tissue	Invasive	Carcass	M	131
	Tissue	Invasive	Carcass	F	132
	Hair	Non-invasive	Damage	F	7
	Blood	Invasive	Capture and release	F	54
	Blood	Invasive	Capture and release	F	129
	Hair	Non-invasive	Buckthorn patches	M	10
	Blood	Invasive	Capture and release	M	66
	Blood	Invasive	Capture and release	M	83
	Blood	Invasive	Capture and release	M	117
*Ursus arctos*	Hair	Non-invasive	Opportunistic	F	F3
*arctos* (*Uaa*)	Hair	Non-invasive	Rub tree	M	M1
	Hair	Non-invasive	Opportunistic	F	F23
	Hair	Non-invasive	Opportunistic	F	F2
	Hair	Non-invasive	Opportunistic	M	JJ5
	Hair	Non-invasive	Opportunistic	F	F12
	Hair	Non-invasive	Hair traps	F	F18
	Tissue	Invasive	Opportunistic	F	F4
	Hair	Non-invasive	Rub tree	M	M6
	Hair	Invasive	Capture and release	F	F15
	Bone	Invasive	Carcass	M	M26
	Hair	Non-invasive	Damage	F	Daniza
	Hair	Non-invasive	Rub tree	M	Gasper
	Hair	Non-invasive	Opportunistic	M	DG2
	Tissue	Invasive	Capture and release	F	DG3
	Hair	Invasive	Capture and release	F	Irma
	Hair	Non-invasive	Hair traps	F	Brenta
	Hair	Non-invasive	Hair traps	F	Kirka
	Feces	Non-invasive	Opportunistic	M	MJ3
	Hair	Non-invasive	Hair traps	M	M4
	Hair	Non-invasive	Hair traps	M	M32
	Hair	Invasive	Opportunistic	M	M33

**Table 2 genes-13-02164-t002:** Genetic diversity parameters and genotyping errors of 13 STR markers (NEW set) of 44 Italian brown bear *U. a. marsicanus* (*Uam)* and *U. a. arctos* (*Uaa)* genotypes. Number of PCR multiplexes, dye of microsatellite primers, allele range, number of alleles (Na), effective number of alleles (Ne), observed (Ho) and unbiased expected (UHe) heterozygosity, Shannon’s information index (I), Hardy–Weinberg equilibrium (HWE), probability of identity for unrelated individuals (PID) or siblings (PIDsib), allelic dropouts (ADO) and false alleles (FA) are included.

Locus	Multiplex	Dye	Allele Range		Na		Ne		Ho		UHe		I		HWE		PID		PIDsib		ADO		FA	
			*Uam*	*Uaa*	*Uam*	*Uaa*	*Uam*	*Uaa*	*Uam*	*Uaa*	*Uam*	*Uaa*	*Uam*	*Uaa*	*Uam*	*Uaa*	*Uam*	*Uaa*	*Uam*	*Uaa*	*Uam*	*Uaa*	*Uam*	*Uaa*
UA03	1	FAM	100–104	96–104	2	3	1.046	2.822	0.045	0.727	0.045	0.661	0.108	1.066	ns	ns	0.91	0.20	0.96	0.48	0.25	0	0	0
UA06	1	HEX	115–119	103–115	2	3	1.046	2.051	0.045	0.455	0.045	0.524	0.108	0.860	ns	ns	0.91	0.31	0.96	0.57	0	0	0	0
UA14	2	HEX	145–161	139–155	2	4	1.963	2.969	0.591	0.727	0.502	0.679	0.684	1.207	ns	ns	0.38	0.17	0.60	0.46	0.17	0	0	0
UA16	2	HEX	122	106–126	1	6	1	4.939	0	0.864	0	0.816	0	1.669	mono	ns	1	0.07	1	0.37	0	0	0	0
UA17	1	FAM	134	138–146	1	3	1	2.568	0	0.545	0	0.625	0	1.018	mono	ns	1	0.22	1	0.50	0	0	0	0
UA25	1	PET	105–125	105–117	4	3	2.696	1.678	0.773	0.318	0.644	0.413	1.136	0.726	ns	ns	0.20	0.39	0.48	0.65	0	0.06	0	0.03
UA51	1	FAM	120–124	112–128	2	5	1.095	2.521	0.091	0.682	0.089	0.617	0.185	1.174	ns	ns	0.84	0.21	0.92	0.50	0	0	0	0
UA57	3	FAM	108–116	112–116	2	2	1.936	1.482	0.545	0.409	0.495	0.333	0.677	0.507	ns	ns	0.38	0.51	0.60	0.71	0.06	0	0	0
UA63	2	NED	114	117–121	1	3	1	1.738	0	0.364	0	0.434	0	0.739	mono	ns	1	0.38	1	0.63	0	0	0	0
UA64	2	PET	113–121	105–109	3	2	2.513	1.713	0.667	0.500	0.617	0.426	1	0.607	ns	ns	0.23	0.43	0.51	0.65	0.04	0.02	0	0
UA65	2	FAM	127–135	123–135	2	4	1.365	2.623	0.318	0.773	0.274	0.633	0.438	1.142	ns	ns	0.57	0.20	0.76	0.49	0.07	0.03	0	0
UA67	3	NED	124–132	124–132	3	3	2.665	1.809	0.762	0.591	0.640	0.458	1.028	0.707	ns	ns	0.22	0.39	0.49	0.62	0.05	0.07	0	0
UA68	3	HEX	129–141	105–137	2	4	1.308	2.942	0.273	0.455	0.241	0.675	0.398	1.150	ns	ns	0.61	0.18	0.79	0.47	0	0	0	0

**Table 3 genes-13-02164-t003:** Details of microsatellite markers compared in this study. The BEST loci selected for long-term monitoring projects for *U. a. marsicanus* (*Uam*) and *U. a. arctos* (*Uaa*) are shown in bold.

Locus	Motif Size	Marker Set		Major Allele Frequency		Gene Diversity (GD)		Polymorphic Information Content (PIC)		Allelic Drop-Out (ADO)		False Alleles (FA)		Probability of Identity (PID)	
		*Uam*	*Uaa*	*Uam*	*Uaa*	*Uam*	*Uaa*	*Uam*	*Uaa*	*Uam*	*Uaa*	*Uam*	*Uaa*	*Uam*	*Uaa*
G10B ^1^	di-	C	-	0.55	-	0.49	-	0.37	-	0.16	-	0	-	0.38	-
G10C ^1^	di-	**C**	C	**0.57**	0.59	**0.49**	0.57	**0.37**	0.51	**0**	0.05	**0**	0.01	**0.38**	0.24
G10H ^1^	di-	-	C	-	0.84	-	0.28	-	0.26	-	0.07	-	0.01	-	0.54
G10L ^1^	di-	C	C	0.69	0.66	0.43	0.49	0.34	0.43	0.05	0.04	0	0	0.39	0.32
G10M ^5^	di-	-	C	-	0.41	-	0.68	-	0.62	-	0.03	-	0.01	-	0.16
G10P ^1^	di-	C	C	0.81	0.30	0.31	0.75	0.26	0.70	0.08	0.05	0	0.25	0.51	0.11
G10X ^1^	di-	-	**C**	-	**0.34**	-	**0.75**	-	**0.71**	-	**0.08**	-	**0**	-	**0.10**
G1D ^1^	di-	**C**	C	**0.53**	0.41	**0.59**	0.75	**0.51**	0.72	**0.09**	0.12	**0**	0	**0.30**	0.10
Mu05 ^2^	di-	C	-	0.81	-	0.31	-	0.26	-	0.06	-	0	-	0.43	-
Mu09 ^2^	di-	-	**C**	-	**0.39**	-	**0.75**	-	**0.72**	-	**0.05**	-	**0**	-	**0.10**
Mu10 ^2^	di-	-	C	-	0.52	-	0.61	-	0.54	-	0.07	-	0.12	-	0.22
Mu11 ^2^	di-	**C**	**C**	**0.74**	**0.41**	**0.41**	**0.73**	**0.36**	**0.68**	**0.07**	**0.01**	**0**	**0**	**0.33**	**0.12**
Mu15 ^2^	di-	C	C	0.97	0.39	0.07	0.73	0.06	0.68	0	0.04	0.04	0.03	0.77	0.12
Mu23 ^2^	di-	-	**C**	-	**0.34**	-	**0.72**	-	**0.68**	-	**0.04**	-	**0**	-	**0.12**
Mu50 ^2^	di-	C	**C**	0.83	**0.36**	0.29	**0.73**	0.24	**0.68**	0	**0.03**	0	**0**	0.51	**0.12**
Mu51 ^2^	di-	**C**	-	**0.62**	-	**0.48**	-	**0.38**	-	**0.01**	-	**0**	-	**0.37**	-
Mu59 ^2^	di-	**C**	C	**0.59**	0.34	**0.49**	0.77	**0.37**	0.73	**0**	0.14	**0**	0	**0.38**	0.09
cxx20 ^3^	di-	C	C	0.46	0.34	0.64	0.73	0.57	0.68	0.09	0.32	0.02	0	0.21	0.12
REN144A06 ^3^	di-	C	-	0.54	-	0.57	-	0.49	-	0.08	-	0.04	-	0.24	-
UA03 ^4^	tetra-	N	**N**	0.95	**0.43**	0.10	**0.65**	0.10	**0.57**	0.25	**0**	0	**0**	0.91	**0.20**
UA06 ^4^	tetra-	N	N	0.95	0.64	0.10	0.51	0.10	0.44	0	0	0	0	0.91	0.31
UA14 ^4^	tetra-	N	**N**	0.59	**0.48**	0.48	**0.66**	0.37	**0.61**	0.17	**0**	0	**0**	0.38	**0.17**
UA16 ^4^	tetra-	N	**N**	mono	**0.25**	mono	**0.80**	mono	**0.77**	mono	**0**	mono	**0**	mono	**0.07**
UA17 ^4^	tetra-	N	**N**	mono	**0.52**	mono	**0.61**	mono	**0.54**	mono	**0**	mono	**0**	mono	**0.22**
UA25 ^4^	tetra-	**N**	N	**0.54**	0.75	**0.63**	0.40	**0.57**	0.37	**0**	0.06	**0**	0.03	**0.20**	0.39
UA51 ^4^	tetra-	N	**N**	0.88	**0.57**	0.22	**0.60**	0.19	**0.55**	0	**0**	0	**0**	0.84	**0.21**
UA57 ^4^	tetra-	**N**	N	**0.59**	0.80	**0.48**	0.33	**0.37**	0.27	**0.06**	0	**0**	0	**0.38**	0.51
UA63 ^4^	tetra-	N	N	mono	0.73	mono	0.42	mono	0.38	mono	0	mono	0	mono	0.38
UA64 ^4^	tetra-	**N**	N	**0.44**	0.70	**0.63**	0.42	**0.55**	0.33	**0.04**	0.02	**0**	0	**0.23**	0.43
UA65 ^4^	tetra-	N	**N**	0.88	**0.55**	0.22	**0.62**	0.19	**0.57**	0.07	**0.03**	0	**0**	0.57	**0.20**
UA67 ^4^	tetra-	**N**	N	**0.48**	0.68	**0.62**	0.45	**0.54**	0.37	**0.05**	0.07	**0**	0	**0.22**	0.39
UA68 ^4^	tetra-	N	**N**	0.93	**0.43**	0.13	**0.66**	0.12	**0.59**	0	**0**	0	**0**	0.61	**0.18**
Mean CURRENT set *				0.67	0.44	0.43	0.67	0.35	0.62	0.05	0.08	0.01	0.03	3.2 × 10^−06^	2.8 × 10^−12^
Mean NEW set *				0.72	0.58	0.36	0.55	0.31	0.49	0.06	0.01	0	0	3.6 × 10^−04^	1.6 × 10^−08^
Mean TOT set *				0.69	0.51	0.40	0.61	0.33	0.56	0.06	0.05	0	0.02	1.1 × 10^−09^	5.2 × 10^−21^
**Mean BEST set ***				**0.57**	**0.42**	**0.53**	**0.69**	**0.45**	**0.64**	**0.04**	**0.02**	**0**	**0**	**2.0 × 10^−05^**	**6.9 × 10^−11^**

^1^ Loci designed using a genomic library of American black bears (*U. americanus*) [55,56,57]. ^2^ Loci designed using a genomic library of European brown bears (*U. arctos*) [58,59]. ^3^ Loci designed using the canid genome [60,61]. ^4^ Loci developed for high-throughput sequencing (HTS) [36]. ^5^ Loci modified from [59]. * Cumulative, not mean, in the case of PID.

**Table 4 genes-13-02164-t004:** Estimates of genetic variability in the four different STR sets compared in this study for *U. a. marsicanus* (*Uam*) and *U. a. arctos* (*Uaa*). The microsatellite panel widely used in Italian monitoring projects (CURRENT), the new set of loci tested in this study (NEW), the complete set of loci (TOT), and the loci with the highest resolution power in individual identification (BEST) are shown. Mean value of observed heterozygosity (Ho), mean value of unbiased expected heterozygosity (UHe), pairs of genotypes that match at all loci (0MM) and at all but 1-2-3 loci (0-1-2-3 MM), probability of identity for unrelated individuals (PID), probability of identity for siblings (PIDsib), effective multiplex ratio (EMR), and marker index (MI) are also listed for *U. a. marsicanus* (*Uam*) and *U. a. arctos* (*Uaa*).

Marker Set *	Ho		UHe		Pairs of Genotypes								PID		PIDsib		EMR		MI	
					0MM		1MM		2MM		3MM									
	*Uam*	*Uaa*	*Uam*	*Uaa*	*Uam*	*Uaa*	*Uam*	*Uaa*	*Uam*	*Uaa*	*Uam*	*Uaa*	*Uam*	*Uaa*	*Uam*	*Uaa*	*Uam*	*Uaa*	*Uam*	*Uaa*
CURRENT	0.463 ± 0.050	0.694 ± 0.048	0.461 ± 0.037	0.685 ± 0.035	0	0	0	0	4.13 × 10^−03^	0	4.13 × 10^−03^	0	3.2 × 10^−06^	2.8 × 10^−12^	1.9 × 10^−03^	1.5 × 10^−05^	1.90	3.34	0.67	2.08
NEW	0.316 ± 0.086	0.570 ± 0.048	0.276 ± 0.074	0.561 ± 0.039	4.13 × 10^−03^	0	6.20 × 10^−03^	0	3.51 × 10^−02^	0	6.40 × 10^−02^	0	3.6 × 10^−04^	1.6 × 10^−08^	2.2 × 10^−02^	3.1 × 10^−04^	2.05	2.45	0.64	1.20
TOT	0.389 ± 0.051	0.636 ± 0.035	0.368 ± 0.045	0.627 ± 0.028	0	0	0	0	0	0	2.07 × 10^−03^	0	1.1 × 10^−09^	5.2 × 10^−21^	4.2 × 10^−05^	2.0 × 10^−09^	1.75	2.92	0.58	1.64
BEST	0.613 ± 0.043	0.716 ± 0.038	0.545 ± 0.023	0.706 ± 0.019	0	0	0	0	1.65 × 10^−02^	0	1.86 × 10^−02^	0	2.0 × 10^−05^	6.9 × 10^−11^	5.3 × 10^−03^	5.4 × 10^−05^	2.18	3.37	0.97	2.09

* *Uam* CURRENT (13 loci): G10B, G10C, G10L, G10P, G1D, Mu05, Mu11, Mu15, Mu50, Mu51, Mu59, cxx20; REN144A06. NEW (13 loci): UA03, UA06, UA14, UA16, UA17, UA25, UA51, UA57, UA63, UA64, UA65, UA67; UA68. TOT (26 loci): G10B, G10C, G10L, G10P, G1D, Mu05, Mu11, Mu50, Mu51, Mu59, cxx20, REN144A06, UA03, UA06, UA14, UA16, UA17, UA25, UA51, UA57, UA63, UA64, UA65, UA67; UA68. BEST (9 loci): G10C, G1D, Mu11, Mu51, Mu59, UA25, UA57, UA64; UA67. * *Uaa* CURRENT (15 loci): G10C, G10H, G10L, G10M, G10P, G10X, G1D, Mu09, Mu10, Mu11, Mu15, Mu23, Mu50, Mu59; cxx20. NEW (13 loci): UA03, UA06, UA14, UA16, UA17, UA25, UA51, UA57, UA63, UA64, UA65, UA67; UA68. TOT (28 loci): G10C, G10H, G10L, G10M, G10P, G10X, G1D, Mu09, Mu10, Mu11, Mu15, Mu23, Mu50, Mu59, cxx20, UA03, UA06, UA14, UA16, UA17, UA25, UA51, UA57, UA63, UA64, UA65, UA67; UA68. BEST (12 loci): G10X, Mu09, Mu11, Mu23, Mu50, UA03, UA14, UA16, UA17, UA51, UA65; UA68.

## Data Availability

The microsatellite genotypes generated and analyzed during the current study are not publicly available due to ongoing work on the genetics of these populations, but are available from the corresponding author upon reasonable request.

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
