# Peer review of "Microsatellite Characterization and Panel Selection for Brown Bear (Ursus arctos) Population Assessment"

_genes, 2022, doi:10.3390/genes13112164_

Round 1
Reviewer 1 Report
This is a nice manuscript addressing a rather technical, but nevertheless important and interesting problem. Authors take on an ambitious task of selecting a minimal reliable STR loci panel that will be sufficient to distinguish Ursus arctos samples regardless of method and quality. Both genetical and statistical porcedures used are appropriate and well described.
My sole suggestion is to add a detailed protocol table with primer sequences, multiplexes and amplification programs all in one place. This will make it much easier for other researchers to follow up on your work and conduct their own studies using your multiplexes and protocols.
Author Response
Dear Editor and Reviewers,
Thank you for giving us the opportunity to submit a revised draft of our manuscript entitled "Microsatellite characterization and panel selection for brown bear (Ursus arctos) population assessment" to Genes. We appreciate the time and effort that you and the reviewers have dedicated to providing your valuable feedback on our manuscript. We are grateful to the reviewers for their insightful comments on our paper. We have been able to incorporate changes to reflect most of the suggestions provided by the reviewers. All the revisions are marked up in the text using the “Track Changes” function.
Here is a point-by-point response to the reviewers’ comments and concerns:
Comments from Reviewer 1
- Comment 1: My sole suggestion is to add a detailed protocol table with primer sequences, multiplexes and amplification programs all in one place. This will make it much easier for other researchers to follow up on your work and conduct their own studies using your multiplexes and protocols.
Response: Thank you for pointing this out. We agree. Therefore, we added Table S1 (see the attachment).
Reviewer 2 Report
The paper demonstrates an important comparison of marker panels to effectively identify individuals and test for genetic diversity in populations. Some comments for minor edits are included below.
Line 78: missing "and" before "to enable ..."
Line 97: 2.1. Sample selection - based on the first sentence of this section it's not clear whether all the samples are being described here or only the samples used for the "NEW panel".
Line 99: Where were the wild bear samples obtained from? (i.e. Italy?). Should link to Figure 1.
Line 117: The Italian Ministry?
Figures 2 and 3: Missing axis titles and scale.
Line 292: "at a global level" does not fit in the sentence.
Line 336-339: "Previous studies" are mentioned but only one is cited ([51]). Add reference to more of the studies being referred to.
Line 348: "non-invasive samples" - but were tissue and blood samples also used?
General notes for Discussion: Avoid the use of run-on sentences and watch comma placement (example lines 294-298). The Discussion could use another sentence or two to clarify the relevance of this study to "isolated" populations, long-term studies and management (these topics were mentioned in Abstract and Discussion but not thoroughly described).
Author Response
Dear Editor and Reviewers,
Thank you for giving us the opportunity to submit a revised draft of our manuscript entitled "Microsatellite characterization and panel selection for brown bear (Ursus arctos) population assessment" to Genes. We appreciate the time and effort that you and the reviewers have dedicated to providing your valuable feedback on our manuscript. We are grateful to the reviewers for their insightful comments on our paper. We have been able to incorporate changes to reflect most of the suggestions provided by the reviewers. All the revisions are marked up in the text using the “Track Changes” function.
Here is a point-by-point response to the reviewers’ comments and concerns:
Comments from Reviewer 2
- Comment 1: Line 78: missing “and” before “to enable ...”
Response: Done (see the manuscript)
- Comment 2: Line 97: 2.1. Sample selection - based on the first sentence of this section it's not clear whether all the samples are being described here or only the samples used for the “NEW panel”.
Response: Agree. We have, accordingly, revised the text for better comprehension.
- Comment 3: Line 99: Where were the wild bear samples obtained from? (i.e. Italy?). Should link to Figure 1.
Response: Agree. We have modified it, detailing the origin of the samples, and linked it to Figure 1.
- Comment 4: Line 117: The Italian Ministry?
Response: We modified in: “Italian Ministry of the Environment”
- Comment 5: Figures 2 and 3: Missing axis titles and scale.
Response: Done. Thank you for this suggestion.
- Comment 6: Line 292: “at a global level” does not fit in the sentence.
Response: Agree. We have, accordingly, revised the text.
- Comment 7: Line 336-339: “Previous studies” are mentioned but only one is cited ([51]). Add reference to more of the studies being referred to.
Response: We changed the sentence to: “Previous studies have shown that on one hand, estimates of heterozygosity based on a few loci provide poor estimates of genome-wide genetic variation, and may not allow the differentiation of individuals, potentially leading to the underestimation of population size [51]. On the other hand, a high number of loci can also have a negative impact on individual identification [70]. Our results confirmed these conclusions.”
- Comment 8: Line 348: “non-invasive samples” - but were tissue and blood samples also used?
Response: Yes. We added “also” in the sentence. We used both invasive and non-invasive sampling but the main relevant data is the application of STR for non-invasive monitoring of the parameters of the species.
- Comment 9: General notes for Discussion: Avoid the use of run-on sentences and watch comma placement (example lines 294-298). The Discussion could use another sentence or two to clarify the relevance of this study to “isolated” populations, long-term studies and management (these topics were mentioned in Abstract and Discussion but not thoroughly described).
Response: Agree. We have, accordingly, modified this section by inserting a new sentence (L.348) referring to the extension of the methods to other isolated populations and for long-term monitoring: “The adopted workflow could be replicated in any small population affected by low genetic variability for which reliable and effective long-term genetic monitoring could be helpful to detect changes in patterns of variability, or confirming the effectiveness of management practices”.
In addition, all spelling and grammatical errors noted by the Reviewers have been corrected.
Reviewer 3 Report
The presented work is very interesting from the point of view of the methodology of developing marker panels that will be used in, for example, conservation genetics. In addition, the new microsatellite panel developed in this way, especially due to the use of tetranucleotide microsatellites, can provide better information on the structure of the population, especially the critically endangered Apennine brown bear.
I just have a question about the wording in the Material and Methods section on line 104 where two types of DNA are mentioned - what types of DNA are the authors referring to? Especially since the rest of the chapter only talks about genomic DNA.
Author Response
Dear Editor and Reviewers,
Thank you for giving us the opportunity to submit a revised draft of our manuscript entitled "Microsatellite characterization and panel selection for brown bear (Ursus arctos) population assessment" to Genes. We appreciate the time and effort that you and the reviewers have dedicated to providing your valuable feedback on our manuscript. We are grateful to the reviewers for their insightful comments on our paper. We have been able to incorporate changes to reflect most of the suggestions provided by the reviewers. All the revisions are marked up in the text using the “Track Changes” function.
Here is a point-by-point response to the reviewers’ comments and concerns:
Comments from Reviewer 3
- Comment 1: I just have a question about the wording in the Material and Methods section on line 104 where two types of DNA are mentioned - what types of DNA are the authors referring to? Especially since the rest of the chapter only talks about genomic DNA.
Response: By ‘genomic DNA’ we mean chromosomal DNA (we used both invasive and non-invasive DNA), in contrast to extrachromosomal DNAs like plasmids, or mitochondrial DNA.